# Differences in Nutrient Intake and Diet Quality among Non-Hispanic Black Adults by Place of Birth and Length of Time in the United States

**DOI:** 10.3390/nu15163644

**Published:** 2023-08-19

**Authors:** Oluwafikayo S. Adeyemi-Benson, Alexandra M. Roehll, Edson Flores, Chelsea R. Singleton

**Affiliations:** 1Department of Kinesiology & Community Health, College of Applied Health Sciences, University of Illinois at Urbana-Champaign, Champaign, IL 61820, USA; aroehll2@illinois.edu; 2Department of Kinesiology and Nutrition, University of Illinois at Chicago, Chicago, IL 60612, USA; flore836@umn.edu; 3Department of Social, Behavioral, and Population Sciences, Tulane School of Public Health and Tropical Medicine, New Orleans, LA 70112, USA; csingle1@tulane.edu

**Keywords:** acculturation, diet quality, nutrient intake, disparities, non-Hispanic Black, NHANES

## Abstract

Prior research suggests that migrating to the United States (US) can negatively affect the diets and health of immigrants. There is limited information on how relocating to the US affects the diets of Black-identifying immigrants. To address this gap, this study examined differences in nutrient intake and diet quality among non-Hispanic Black adults by place of birth and length of time in the US. Cross-sectional data from the National Health and Nutrition Examination Survey (2005–2016) were analyzed. Approximately 6508 non-Hispanic Black adults were categorized into three groups: foreign-born (FB) living in the US <10 years (*n* = 167), FB living in the US ≥ 10 years (*n* = 493), and US-born (*n* = 5848). Multivariable-adjusted logistic and linear regression models were evaluated to identify differences in nutrient intake and diet quality (as measured by the Healthy Eating Index (HEI) of 2015) across the three groups when controlling for socio-demographics. Compared to US-born adults, both FB groups had significantly higher HEI-2015 scores and higher odds of meeting dietary recommendations for several nutrients: saturated fat, sodium, and cholesterol. There were no differences in nutrient intake between the two FB groups; however, FB (<10 years) adults had better diet quality than FB (≥10 years) ones. Place of birth and length of time in the US were associated with dietary intake among non-Hispanic Black adults. More research is needed to improve understanding of dietary acculturation among Black-identifying immigrants in the US.

## 1. Introduction

A healthy diet and lifestyle are essential to chronic disease prevention [1]. While dietary preferences and habits can vary substantially between people with different cultural backgrounds, most Americans’ diets exceed the recommended intake for saturated fats, sodium, added sugars, and refined grains [2]. Poor diet is a known risk factor for several chronic diseases, including obesity, cardiovascular disease (CVD), strokes, cancer, and type 2 diabetes [3]. Recognizing how poor diet quality and nutrient intake affect the health status of racial/ethnic minorities is an important public health priority in the United States (US) [4]. Despite recently documented improvements to the quality of Americans’ diets, not every subpopulation has benefitted [5]. Non-Hispanic Black adults have experienced the least improvement among all racial/ethnic groups [5]. Furthermore, previous studies found that Black adults have less-favorable nutrient intakes, lower adherence to dietary guidelines, and poorer dietary quality compared to their White counterparts [5,6]. These nutritional inequities have great potential to further exacerbate disparities in chronic disease risk by race/ethnicity in the US [4]. 

Unfortunately, the field’s understanding of the diets of Black-identifying populations in the US is limited in scope. Currently, there is limited understanding of differences in dietary practices and preferences among Black adults given their culture and lived experiences. The recent immigration wave of people who self-identify as non-Hispanic Black (i.e., people from African, Caribbean, Central American, or South American nations) underscores the need to expand understanding of the diets of Black adults and children in the US [7,8]. Between 2000 and 2013, the number of Black immigrants in the US increased by 56%, with migration from Africa increasing by 137% [7]. In 2017, there were an estimated four million Caribbean immigrants living in the US [8]. 

Moving to the US can result in dietary acculturation, which entails changes to an individual’s traditional diet that result in alignment with the typical American diet [9]. In general, dietary acculturation has been found to have detrimental effects on the diets of immigrants, which consequently can increase the risk of diet-related chronic diseases among immigrant populations [9]. For example, a prior study found that adapting to the US lifestyle was associated with the loss of cultural culinary preferences and increased the consumption of unhealthy foods among immigrants despite improvements in their socioeconomic status [10]. Several studies have linked acculturation measures to changes in dietary intake in several immigrant populations, including Puerto Rican, South Asian, and Filipino adults [11,12,13]. Overall, findings from the literature support the hypothesis that relocating to the US can result in significant declines in diet quality. 

Given the scarcity of scientific research on the diets of Black-identifying immigrant populations and the growing number of Black immigrants in the US, there is a need to study the differences in nutrient intake and diet quality by place of birth and length of time in the US among Black adults. A prior study reported disparities in diet quality between US-born and foreign-born Black adults, with the former having poorer diet quality [14]. However, the study did not examine diet in relation to the 2015–2020 Dietary Guidelines for Americans (DGAs). Thus, this study aimed to examine differences in nutrient intake and diet quality between US-born and foreign-born (henceforth, FB) non-Hispanic Black adults who participated in the National Health and Nutrition Examination Survey (NHANES). In addition, this study evaluated the role of length of time in the US by examining differences between FB Black adults who migrated to the US fewer than 10 years ago and those who migrated more than 10 years ago. It was hypothesized that FB Black adults (<10 years) would have better diet quality than US-born Black adults; however, FB Black adults (≥10 years) would have diet profiles similar to US-born Black adults. When comparing the FB groups, FB Black adults (<10 years) were expected to have better diet quality than FB Black adults (≥10 years).

## 2. Materials and Methods

### 2.1. Data Source

Cross-sectional data collected from participants of the Centers for Disease Control and Prevention’s National Health and Nutrition Examination Survey (NHANES) cycles in 2005–2016 were obtained and analyzed. NHANES collects data from a multistage, stratified probability-cluster sample of the non-institutionalized U.S. population [15]. Data on nutrition and health are collected from participants by conducting a series of interviewer-administered questionnaires and physical examinations [15]. A total of 60,936 adults and children participated in the six selected cycles. Individuals who did not self-identify as non-Hispanic Black and were less than <20 years of age (*n* = 53,863) were excluded from this study, which left 7073 non-Hispanic Black adults. Participants with missing day one 24 h recall data were also excluded (*n* = 565). Thus, the analytical sample for this study comprised 6508 non-Hispanic Black adults. 

Measures representing place of birth (US vs. other) and length of time in the U.S. were used to categorize participants into three distinct groups: FB Black adults who migrated fewer than 10 years ago (*n* = 167; 3.0%), FB Black adults who migrated more than 10 years ago (*n* = 493; 7.2%), and US-born Black adults (*n* = 5848; 89.8%). NHANES collects self-reported information about place of birth and length of time in the US [15]. These two measures are often used as proxy measures of acculturation in studies on the health and health behaviors of immigrant populations in the US [11,12,13,14]. Since NHANES does provide separate race and ethnicity data, Black-identifying Hispanics could not be separated from other Hispanic adults. Therefore, the current study only included non-Hispanic Black adults. The National Center for Health Statistics Institutional Review Board (IRB) approved NHANES, and all participants provided written informed consent [15]. The IRB at the University of Illinois at Urbana-Champaign deemed this research exempt.

### 2.2. Nutrient Intake

Nutrient intake data were examined for all participants included in the analytical sample. The dietary intake interview of NHANES, titled “What We Eat in America”, was conducted in partnership with the U.S. Department of Agriculture using a computerized data collection instrument [16]. Each participant was eligible for two days of 24 h recall; the first day was conducted in-person during the initial NHANES interview, while the second day was conducted over the telephone approximately 3–10 days later [16]. As stated above, 6508 non-Hispanic Black adults participating in NHANES 2005–2016 had complete dietary data for the first day; 4867 (69%) had complete data for the second day. Only data from the first day were analyzed since 31% of the sample did not complete the second 24 h dietary recall. 

Measures examined included total energy (kcal per day), protein (grams per day), carbohydrates (grams per day), total sugar (grams per day), dietary fiber (grams per day), total fat (grams per day), saturated fat (grams per day), cholesterol (milligrams per day), and sodium (milligrams per day). To identify participants who met recommendations for nutrient intake, participants’ consumption levels for each nutrient were compared to the recommended level of intake mentioned in the 2015–2020 Dietary Guidelines for Americans (DGAs) [17]. According to the DGAs, the following are the recommended intake range(s): 20–35% of energy from total fat, 10–35% of energy from protein, 45–65% of energy from carbohydrates, 14 g/1000 kcal/day from fiber, <10% of energy from saturated fat, and <2300 mg/day of sodium [17]. The 2015–2020 DGAs do not have a recommended consumption amount for total sugars and cholesterol. However, the World Health Organization (WHO) recommends that <5% of energy intake should come from added sugar. Thus, total sugar intake was compared to this recommendation to identify the proportion of FB Black adults and U.S.-born Black adults who had a total sugar intake amount of <5% of energy intake [18]. As for the daily recommendation for cholesterol intake, the goal of <300 mg/day was utilized. This goal was included in the prior iteration of the DGAs (2010–2015) [17].

### 2.3. Diet Quality

The Healthy Eating Index (HEI) is a diet quality index that measures an individual or population’s dietary alignment with the DGAs [19,20]. It can be used to assess the conformance of any meal or group of foods to the diet recommendations outlined in the DGAs [19,20]. For the current study, study participants’ (*n* = 6508) day one 24 h recall data were analyzed using the simple HEI scoring algorithm to generate HEI-2015 total and component scores [21]. Since the simple HEI scoring algorithm was used, the HEI-2015 scores presented in this study do not represent usual intake (i.e., long-term intake). Rather, they represent an estimation of how each participant’s consumption on day one of the dietary interview aligned with the DGAs. 

The HEI-2015 total score ranges from 0 to 100, with 100 indicating perfect alignment. It consists of 13 components, including total fruits, whole fruits, total vegetables, greens and beans, total protein foods, seafood and plant proteins, whole grains, dairy, fatty acids, refined grains, sodium, added sugars, and saturated fats. Total fruits, whole fruits, total protein foods, total vegetables, seafood and plant proteins, and greens and beans all contribute five points each to the total score. The other dietary components all contribute ten points to the total score [19]. Refined grains, sodium, added sugars, and saturated fats are considered measures of moderation; higher consumption of these foods will lower the HEI total score. All others are considered measures of adequacy, so higher consumption of these items will increase HEI total score.

### 2.4. Other Measures

In addition to nutrient intake and diet quality, the following measures were examined: current age (years), sex (male vs. female), education level (<high school diploma, high school diploma or equivalent, some college, or ≥college degree), marital status (married vs. other), number of people living in the household, poverty-to-income ratio (PIR), and day of the week the 24 h recall interview was completed. These socio-demographic measures were obtained via the interviewer-administered questionnaires. NHANES investigators estimated each participant’s PIR from their self-reported annual income. The PIR represents the ratio of an individual’s annual household income to the federal poverty level for their household size the year the NHANES interview was conducted [22]. Days of the week for the recall interview were categorized to compare participants who had their interviews on the weekend (Friday–Sunday) to participants who had interviews on a weekday (Monday–Thursday).

### 2.5. Statistical Analysis

To examine the characteristics of the analytical sample, descriptive statistics were calculated (i.e., weighted means and frequencies). Analysis of variance (ANOVA) and chi-square tests were used to identify differences in socio-demographic measures across the three groups representing acculturation: FB Black adults (<10 years), FB Black adults (≥10 years), and U.S.-born Black adults. The weighted mean intake of each nutrient among the three groups and the weighted percentage of participants in the groups who met the dietary recommendations for each nutrient were calculated. Logistic regression was used to determine if the odds of meeting dietary recommendations for nutrient intake were significantly different among the three acculturation groups. Models were run to compare (1) both groups of FB Black adults to US-born Black adults and (2) FB Black adults (<10 years) to FB Black adults (≥10 years).

Linear regressions were used to examine the association differences in HEI-2015 total scores between the three acculturation groups and both groups of FB Black adults. The unadjusted model included only the variables representing the three groups. The adjusted model also included the variables of age, sex, education level, marital status, PIR, number of household members, and day of the week of the 24 h recall interview. These socio-demographic variables were included because prior research has shown they are associated with dietary intake [23]. Confidence intervals that did not include the null value of 1.0 and had *p* values < 0.05 were considered statistically significant. All analyses were conducted by using SAS version 9.4 [24]. Since NHANES employs a complex sampling scheme, appropriate sampling weights were applied to the descriptive statistics and regression analyses.

## 3. Results

Descriptive statistics stratified by the three groups are presented in Table 1. Among the 6508 non-Hispanic Black adults, the mean age was 44.6, 44.3% were male, and 17.6% had ≥college degree. Most of the study participants (65.8%) reported a marital status other than “married”. Participants had three household members on average and a poverty-to-income ratio of 2.3. Significant demographic differences were observed across the three groups for every measure of interest except the PIRs. A higher percentage of FB Black adults (≥10 years) had ≥college degree compared to the FB Black adults (<10 years) and US-born Black adults, and a higher percentage of FB Black adults (<10 years) reported being married compared to the other two groups.

Descriptive information on nutrient intake is displayed in Table 2. Less than 3% of participants in all three groups met the intake recommendations for dietary fiber, and less than 4% met recommendations for total sugar intake. Less than 16% of participants in all three groups met the intake recommendation for sodium. While ≥60% of foreign-born adults met intake recommendations for saturated fat, only 43% of US-born adults met the saturated fat recommendation.

Results from the logistic regression models examining associations between foreign-born status, length of time in the US, and odds of meeting recommendations for nutrient intake are displayed in Table 3. The unadjusted logistic regression models indicated that FB Black adults (<10 years) had greater odds of meeting the dietary recommendations for saturated fat and carbohydrates than US-born Black adults. FB Black adults (≥10 years) had greater odds of meeting the dietary recommendations for total fat, saturated fat, protein, and cholesterol compared to US-born Black adults. After adjusting for age, sex, education level, marital status, PIR, number of household members, and day of the week, FB Black adults (<10 years) had significantly higher odds of meeting the dietary recommendations for saturated fat (odds ratio (OR), 2.74; 95% CI, 1.62–4.63), protein (OR, 1.84; 95% CI, 1.01–3.34), carbohydrates (OR, 1.71; 95% CI, 1.17–2.49), cholesterol (OR, 1.71; 95% CI, 1.16–2.52), and sodium (OR, 2.19, 95% CI = 1.16–4.14) than US-born Black adults. FB Black adults (≥10 years) had significantly higher odds of meeting dietary recommendations for total fat (OR, 1.65; 95% CI, 1.34–2.04), saturated fat (OR, 2.24; 95% CI, 1.77–2.83), cholesterol (OR, 1.56, 95% CI, 1.22–1.99), and sodium (OR, 1.57; 95% CI, 1.11–2.20) than US-born Black adults. When comparing the two groups of FB Black adults, no significant differences in the odds of meeting nutrient intake recommendations were detected, except for carbohydrates. After adjusting for covariates, FB Black adults (<10 years) had higher odds of meeting the carbohydrate recommendation (OR, 1.63; 95% CI, 1.24–2.13). 

Table 4 presents results from linear regression models assessing associations between foreign-born status, length of time in the US, and HEI-2015 total score. The mean HEI-2015 total scores were 54.6 for FB Black adults (<10 years), 56.3 for FB Black adults (≥10 years), and 47.8 for US-born Black adults. Compared to US-born Black adults, FB Black adults (≥10 years) [β = 8.5; standard error [SE] = 0.7; *p* = 0.0001] and FB Black adults (<10 years) [β = 6.8; SE = 1.2; *p* = 0.0001] had higher HEI-2015 total scores. The significance was retained after adjusting for age, sex, education level, marital status, PIR, number of household members, and day of the week. When comparing the groups of FB Black adults, the unadjusted model yielded no significant results. However, the adjusted model indicated that FB adults (<10 years) had a higher HEI-2015 total scores than FB (≥10 years) after adjusting for all covariates [β = 2.6; SE = 0.7; *p* = 0.0004].

## 4. Discussion

This study aimed to determine if acculturation, place of birth, and length of time in the US are associated with nutrient intake and diet quality among non-Hispanic Black adults who participated in NHANES. It was hypothesized that FB Black adults who migrated to America fewer than 10 years ago would have better diet quality than US-born Black adults and FB Black adults who migrated 10 years ago or more. Overall, the findings from this study supported the hypothesis that FB Black adults (<10 years) had better diet quality than US-born Black adults. However, FB Black adults who migrated more than 10 years ago also had better diet quality than US-born Black adults. This finding does not align with the hypothesis that FB Black adults who have been in the US for more than 10 years would have diets similar to US-born Black adults. It appears that FB Black adults, regardless of their length of time in the US, had better diets than US-born Black adults. When comparing the FB groups, the odds of meeting nutrient recommendations were similar between the groups; however, the estimates from the linear regression model revealed that FB Black adults (<10 years) had slightly higher diet quality scores than FB Black adults (≥10 years). 

Unlike studies that focused on Latino and Asian immigrants [11,12,13,25,26,27], FB Black adults had better diet quality than US-born ones, regardless of the year they migrated to America. This finding aligns with results from a study by Brown et al., which reported that being foreign-born is associated with significantly higher diet quality scores (as measured by the Alternative HEI-2010 and DASH scores) and greater intake of healthier foods (e.g., fruits, vegetables) among Black adults in the US [14]. Brown et al. also concluded that diet quality did vary significantly by length of time in the US among FB Black adults. Thus, it is possible that length of time in the US is not associated with dietary intake among foreign-born adults who self-identify as non-Hispanic Black in the same manner as immigrants from other regions of the world. 

Evidence from qualitative studies provides more in-depth information on the relationship between the measures that represent acculturation and dietary intake among immigrants of African descent. Paxton et al. found that West-African immigrants living in New York, NY, reported strong efforts to maintain their traditional diets over time, which typically comprised fruits, vegetables, and grains [28]. However, they found it difficult to maintain this diet in their new environment. The participants did see evidence of dietary acculturation among their children [28], which aligns with findings from a study by Jakub et al. [29]. Jakub et al. discovered that the children of African immigrants had diets closer in profile to American youth and were more influenced by their peers and environment [29]. It is possible that Black adults who migrate to the US try hard to maintain their traditional diets over time, and the effects of dietary acculturation are more evident in their children. Given the limited number of quantitative studies on this topic, additional research is needed to confirm these findings and connect behavioral factors (e.g., cooking practices, food-purchasing habits, food preferences, etc.) to dietary outcomes among FB Black adults and their children. 

As previously mentioned, prior studies have linked measures that represent acculturation, such as length of time in the US, to poorer dietary quality among immigrant populations [11,12,13,25,26,27]. A study by Thomson et al. reported that acculturation was associated with poorer dietary quality and higher body mass indexes among Mexican immigrants in the US [25]. It is likely that acculturation influences diet and health differently across immigrant populations in the US. Greater emphasis and study should be devoted to assessing these differences and their connection to racial/ethnic disparities in dietary behavior and chronic disease risk. 

Overall, it is important to note that all three groups had large proportions of individuals who were not meeting national nutrient recommendations. For example, a small percentage of participants in all three groups met recommendations for intake of dietary fiber, total sugars, and sodium. These findings align with evidence from population-based studies of nutrient intake and adherence to dietary recommendations that focused on non-Hispanic Black adults [5,6,23,30]. A study by Thompson et al., which examined differences in nutrient intake between non-Hispanic White and Black men living in the U.S., found that less than 5% of men met the recommendations for dietary fiber and total sugar intake [30]. Furthermore, a recent “What We Eat in America” assessment of usual intake among non-Hispanic Black adults reported that most Black adults in the US surpass national recommendations for sodium intake [31]. Meeting national recommendations for nutrient intake is important, as scientific evidence indicates strong associations between saturated fat, dietary cholesterol, sodium, and CVD [32]. Since Black Americans experience high prevalence rates of many CVD risk factors (e.g., obesity, metabolic syndrome, type 2 diabetes, and high blood pressure), it is important that the field identifies factors that influence dietary intake, such as acculturation [7,33,34,35].

### Strengths and Limitations

This study has strengths and limitations. Use of the nationwide NHANES dataset was a strength because it included a large, diverse sample of non-Hispanic Black adults. In addition, use of HEI-2015 was a strength because it directly measured how an individual’s diet aligned with the DGAs. Key limitations included the low sample size for FB Black adults (<10 years), which might have affected ability to observe statistically significant findings for some dietary measures. This study employed a cross-sectional design, so causal associations could not be studied. Because a significant number of study participants had missing data for the second 24 h recall interview, only data from the first recall interview were analyzed. Thus, HEI-2015 scores reflecting usual intake were not calculated. All findings on nutrient intake and diet quality solely reflect the consumption reported by participants on the first day of the dietary interview. As previously mentioned, the inability to examine Black-identifying Hispanic adults was a limitation. Individuals included in the analytical sample solely reflect non-Hispanic Black adults living in the US. Future studies should include Black Hispanic adults, which likely includes individuals from Latin America and the Caribbean with diverse cultural backgrounds.

The primary independent variables (i.e., foreign-born status and length of time in the US) were a major limitation of this study for two key reasons. First, these variables are only proxy measures of acculturation. Although used in prior research, they do not capture the full extent and experience of acculturation among immigrant populations [14]. Future studies should use a validated acculturation scale tailored to the target population of interest. Second, these variables only permit a simplistic comparison of foreign-born to US-born Black-identifying adults, which does not capture the generational effects associated with immigration. Studies have found intergenerational differences in dietary change among West-African immigrants, with first-generation West-African adults exhibiting more dietary acculturation compared to their immigrant parents [36]. NHANES data do not provide data to determine their generational status. In addition, the data source does not have information on ancestry, cultural beliefs, family dynamics (e.g., gender roles, cooking behaviors), or relevant environmental factors (e.g., urban/rural status, access to healthy food retailers, etc.). Having this information would have facilitated a more in-depth analysis of dietary differences between foreign-born and US-born Black adults that accounted for the complexity of these associations and the historical diversity of these groups. Future studies should consider these limitations and conduct qualitative and quantitative research that addresses these gaps in knowledge. 

## 5. Conclusions

In summary, FB Black adults had higher odds of meeting several nutrient recommendations and had better diet quality compared to US-born Black adults, regardless of their length of time in the US. Understanding the similarities and differences among these groups is valuable for developing tailored dietary and lifestyle interventions and decreasing the risk of diet-related chronic diseases among non-Hispanic Black adults in the US. The lived experience of Black-identifying adults that migrate to US should be studied in relation to dietary intake. Although length of time in the US appears to not be a salient factor, there may be other factors that may be relevant the dietary behaviors of foreign-born Black adults: stress, underemployment, racial discrimination, economic expectations from family/community back home. Overall, this study contributes to the bodies of knowledge about the diets of immigrant populations and differences in the diets of adult immigrants who self-identify as Black in the US. This study provides valuable knowledge to the field on diet quality and nutrient intake among non-Hispanic Black immigrants. The results may be useful to nutrition educators and practitioners working to improve the health of this minority population. Additional studies are needed to explore the importance of factors contributing to changes in diet due to acculturation and their overall impact on the health and health behaviors of immigrants who self-identify as non-Hispanic Black in the US.

## Figures and Tables

**Table 1 nutrients-15-03644-t001:** Demographic characteristics of study participants stratified by foreign-born status and length of time in the US, *n* (%) or mean (±SE) ^a,b,c^.

Characteristic:	All Participants *n* = 6508 ^d,e^	Foreign-Born (<10 years) 167 (3.0%)	Foreign-Born (≥10 years) 493 (7.23%)	US-Born 5848 (89.76%)	*p* Value ^f^
Age, years	44.6 (0.3)	36.5(1.1)	47.4 (0.7)	44.6 (0.4)	0.0006
Sex:					0.002
Male	3123 (44.3)	96 (56.9)	249 (46.6)	2778 (43.7)	
Female	3385 (55.7)	71 (43.1)	244 (53.4)	3070 (56.3)	
Education Level:					0.0001
<High School	1569 (21.6)	33 (19.2)	98 (16.2)	1438 (22.1)	
HS or Equivalent	1687 (25.8)	34 (22.5)	92 (18.1)	1561 (26.5)	
Some College	2159 (35.0)	53 (30.9)	167 (36.1)	1939 (35.0)	
≥College Degree	1087 (17.6)	47 (27.5)	134 (29.6)	906 (16.3)	
Marital Status:					0.0001
Married	2350 (34.2)	89 (51.8)	250 (48.8)	2011 (32.5)	
Other	4154 (65.8)	78 (48.2)	243 (51.2)	3833 (67.5)	
Number of Household Members	3.1 (0.0)	3.6 (0.2)	3.2 (0.1)	3.1 (0.0)	0.0001
Poverty-to-Income Ratio	2.3 (0.1)	2.1 (0.1)	2.7 (0.1)	2.3 (0.1)	0.11

^a^ NHANES, National Health and Nutrition Examination Survey. ^b^ SE, Standard Error. ^c^ Data Source: National Health and Nutrition Examination Survey. ^d^ Cell counts may not total to sample size because of missing data. ^e^ Percentages may not total to 100 because of rounding. ^f^
*p* values determined by χ^2^ test or analysis of variance (ANOVA), with *p* < 0.05 considered statistically significant.

**Table 2 nutrients-15-03644-t002:** Descriptive information on nutrient intake by foreign-born status and length of time in the US ^a^.

Nutrient:	Mean Intake (SE) ^b^	Recommended Intake	% Meeting Guideline
**Total Energy Intake, kcal**			
Foreign Born (<10 years)	1883 (65)	—	—
Foreign Born (≥10 years)	1876 (63)		
US Born	2164 (21)		
**Total Fat, g ^c^**			
Foreign Born (<10 years)	61.9 (3.4)	20–35% of energy	51.1
Foreign Born (≥10 years)	64.7 (2.7)		56.3
US Born	84.4 (0.1)		44.7
**Saturated Fat, g**			
Foreign Born (<10 years)	20.0 (1.2)	<10% of energy	66.4
Foreign Born (≥10 years)	20.3 (1.0)		64.1
US Born	26.6 (0.3)		43.2
**Protein, g**			
Foreign Born (<10 years)	77.9 (2.8)	10–35% of energy	91.4
Foreign Born (≥10 years)	77.1 (2.2)		90.3
US Born	80.0 (0.8)		85.9
**Dietary Fiber, g**			
Foreign Born (<10 years)	15.8 (0.9)	14 g/1000 kcal	1.8
Foreign Born (≥10 years)	16.8 (0.5)		2.9
US Born	13.8 (0.2)		2.3
**Carbohydrates, g**			
Foreign Born (<10 years)	252.2 (9.2)	45–65% of energy	65.9
Foreign Born (≥10 years)	241.9 (8.7)		60.3
US Born	257.5 (2.6)		55.4
**Total Sugars, g**			
Foreign Born (<10 years)	96.2 (4.6)	<5% of energy ^d^	3.4
Foreign Born (≥10 years)	101.4 (4.2)		3.1
US Born	123.0 (1.7)		3.6
**Cholesterol, mg**			
Foreign Born (<10 years)	254.6 (16.0)	<300 mg ^e^	68.3
Foreign Born (≥10 years)	259.4 (13.6)		70.9
US Born	314.4 (3.7)		60.1
**Sodium, mg**			
Foreign Born (<10 years)	3011.9 (140.0)	<2300 mg ^e^	15.2
Foreign Born (≥10 years)	2975.0 (107.6)		13.6
US Born	3444.5 (33.0)		10.5

^a^ NHANES, National Health and Nutrition Examination Survey. ^b^ SE, standard error. ^c^ g, grams. ^d^ Recommended intake of added sugars from the World Health Organization. ^e^ mg, milligrams.

**Table 3 nutrients-15-03644-t003:** Results from logistic regression models examining associations between foreign-born status, length of time in the U.S., and odds of meeting national recommendations for nutrient intake ^a^.

Nutrient:	Odds Ratio (95% CI) ^b^		Odds Ratio (95% CI)	
	Unadjusted ^c^	Adjusted ^d^	Unadjusted ^c^	Adjusted ^d^
**Total Energy Intake**				
Foreign Born (<10 years)	—	—	—	—
Foreign Born (≥10 years)				
US Born				
**Total Fat**				
Foreign Born (<10 years)	1.29 (0.87–1.92)	1.49 (0.93– 2.38)	0.81 (0.57–1.16)	1.11 (0.74–1.67)
Foreign Born (≥10 years)	1.60 (1.33–1.91)	1.65 (1.34–2.04)	REF	REF
US Born	REF ^e^	REF	-	-
**Saturated Fat**				
Foreign Born (<10 years)	2.60 (1.58–4.26)	2.74 (1.62–4.63)	1.10 (0.73–1.66)	1.33(0.93–1.89)
Foreign Born (≥10 years)	2.35 (1.88–2.94)	2.24 (1.77–2.83)	REF	REF
US Born	REF	REF	-	-
**Protein**				
Foreign Born (<10 years)	1.76 (0.97–3.19)	1.84 (1.01–3.34)	1.14 (0.66–1.99)	1.14 (0.73–1.77)
Foreign Born (≥10 years)	1.54 (1.08–2.20)	1.36 (0.94–1.96)	REF	REF
US Born	REF	REF	-	-
**Dietary Fiber**				
Foreign Born (<10 years)	0.79 (0.29–2.17)	0.93 (0.32–2.71)	0.62 (0.19–2.01)	0.78 (0.17–3.53)
Foreign Born (≥10 years)	1.27 (0.63–2.55)	1.27 (0.60–2.74)	REF	REF
US Born	REF	REF	-	-
**Carbohydrates**				
Foreign Born (<10 years)	1.56 (1.10–2.19)	1.71 (1.17–2.49)	1.27 (0.94–1.73)	1.63 (1.24–2.13)
Foreign Born (≥10 years)	1.22 (0.98–1.53)	1.19 (0.95–1.50)	REF	REF
US Born	REF	REF	-	-
**Total Sugars**				
Foreign Born (<10 years)	0.95 (0.31–2.92)	1.10 (0.33–3.66)	1.09 (0.35–3.37)	0.90 (0.27–3.09)
Foreign Born (≥10 years)	0.88 (0.44–1.76)	1.07 (0.54–2.14)	REF	REF
US Born	REF	REF	-	-
**Cholesterol**				
Foreign Born (<10 years)	1.43 (0.96–2.13)	1.71 (1.16–2.52)	0.89 (0.59–1.32)	0.95 (0.68–1.32)
Foreign Born (≥10 years)	1.62 (1.29–2.03)	1.56 (1.22–1.99)	REF	REF
US Born	REF	REF	-	-
**Sodium**				
Foreign Born (<10 years)	1.52 (0.92–2.51)	2.19 (1.16–4.14)	1.14 (0.68–1.92)	1.32 (0.77–2.27)
Foreign Born (≥10 years)	1.33 (0.96–1.84)	1.57 (1.11–2.20)	REF	REF
US Born	REF	REF	-	-

^a^ NHANES, National Health and Nutrition Examination Survey. ^b^ CI, confidence interval. ^c^ The unadjusted model with only the variable(s) representing foreign-born status and/or length of time in the US. ^d^ Model adjusted for age, sex, education level, marital status, poverty level, number of household members, and day of the week. ^e^ REF, reference group.

**Table 4 nutrients-15-03644-t004:** Results from linear regression models examining the association between foreign-born status, length of time in the US, and HEI-2015 total score ^a,b^.

Group:	Mean Score (SE)	β (SE) [*p* Value] ^c,d^		β (SE) [*p* Value]	
		Unadjusted ^e^	Adjusted ^f^	Unadjusted ^e^	Adjusted ^f^
Foreign Born (<10 years)	54.6 (1.2)	6.8 (1.2) [<0.0001]	8.9 (1.3) [<0.0001]	−1.7 (1.4) [0.23]	2.6 (0.7) [0.004]
Foreign Born (≥10 years)	56.3 (0.7)	8.5 (0.7) [<0.0001]	7.3 (0.7) [<0.0001]	REF	REF
US Born	47.8 (0.2)	REF ^g^	REF	-	-

^a^ HEI, Healthy Eating Index. ^b^ NHANES, National Health and Nutrition Examination Survey. ^c^ Coefficient estimate. ^d^ SE, standard error. ^e^ The unadjusted model with only the variable(s) representing foreign-born status and/or length of time in the US. ^f^ Model adjusted for age, sex, education level, marital status, poverty level, number of household members, and day of the week. ^g^ REF, reference group.

## Data Availability

The data used for this research project are publicly available on the website for the National Health and Nutrition Examination Survey (NHANES): https://www.cdc.gov/nchs/nhanes/index.htm. The data were accessed on 15 December 2019.

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
