# Peer review of "Differences in Nutrient Intake and Diet Quality among Non-Hispanic Black Adults by Place of Birth and Length of Time in the United States"

_nutrients, 2023, doi:10.3390/nu15163644_

Round 1

Reviewer 1 Report

Differences in Nutrient Intake and Diet Quality among Non-Hispanic Black Adults by Place of Birth and Length of Time in the United States

Thank you for allowing me to review this manuscript which examined differences in nutrient intake and diet quality among non-Hispanic Black adults by place of birth and length of time in the US, using cross-sectional data from the National Health and Nutrition Examination Survey (2005- 2016).

·         All but 1 reference is from 2020 or before, suggest doing an updated literature search to make sure to include recent contributions.

·         Suggest clarifying the decision not to use NHANES weights, results presented refer only to the NHANES sample and are not generalizable to the full US population.

·         The comparison of FB to US non-Hispanic Black adults born is overly simplistic and warrants greater discussion in the limitations section.

Materials and Methods

·         Data source is very well explained.

·         Please explain the decision to use only the surveys and not the lab results from the mobile examination center (MEC)

o   Data on nutrition and health are collected from participants by conducting a series of interviewer-administered questionnaires and physical examinations

·         How was the cut-off of < or > 10 years created? For FB<10yr n=167 across 11 years means approximately 15 per year.

o   FB Black adults who migrated less than 10 years ago (n = 167; 3.0%), FB Black adults who migrated greater than 97 10 years ago (n = 493; 7.2%), and US-born Black adults (n = 5,848; 89.8%).

·         While it’s understandable the authors used the data available, something about the comparison to FB vs US born seems overly simplistic. 

o   If a foreign-born person immigrated with parents, are they the one’s maintaining their traditional diet? If a husband comes with his wife, is she the one shopping for or cooking traditional foods? Would a single person maintain a traditional diet?

o   Would a US born Black adult be aware of ways to achieve a diet similar to FB counterparts? Would that have been modeled in the home for them just because they share skin color?

o   Are all US born Blacks direct descendants from FB countries represented?  What about the cultural/societal impact of diet related to mixed races and slavery?

Results – For Table 2 – indicate within the table if there was a significant difference between groups.

Author Response

Reviewer 1:

Differences in Nutrient Intake and Diet Quality among Non-Hispanic Black Adults by Place of Birth and Length of Time in the United States

Thank you for allowing me to review this manuscript which examined differences in nutrient intake and diet quality among non-Hispanic Black adults by place of birth and length of time in the US, using cross-sectional data from the National Health and Nutrition Examination Survey (2005- 2016).

--Special thanks to the reviewer for identifying areas of improvement for our research paper. We appreciate you providing us with detailed comments and instructions. I invite you to view the highlighted changes and let us know if they are satisfactory. Our detailed responses to your comments are below in bold text.

All but 1 reference is from 2020 or before, suggest doing an updated literature search to make sure to include recent contributions.

--Thank you for this comment. We agree that we should update our literature review. We replaced several of our citations with more recent contributions to let readers know that the points we make throughout the manuscript reflect the current state of science.

Suggest clarifying the decision not to use NHANES weights, results presented refer only to the NHANES sample and are not generalizable to the full US population.

--Lines 189-190: Thank you for asking for clarification. It is important to acknowledge that we indeed utilized the NHANES sample weight in our study; this information is provided to readers on lines 189-190 in the Methods.

--Lines 340-344: In addition, we wish to emphasize that the scope of this analysis was limited to non-Hispanic Black identifying participants of NHANES. Thus, the analytical sample does not reflect the entirety of the US population. Instead, it best represents the Black-identifying population residing in America. We make this point in the limitations section of the Discussion.

The comparison of FB to US non-Hispanic Black adults born is overly simplistic and warrants greater discussion in the limitations section.

--Thank you for your valuable feedback. We completely agree that the comparison of Facebook (FB) to US non-Hispanic Black adults born is indeed a complex topic that deserves a more comprehensive discussion in the limitations section. In the limitations section, we provided a more in-depth explanation of the reasons for choosing this particular comparison and acknowledged its potential drawbacks. We addressed factors such as socio-economic status, cultural differences, historical context, and other relevant variables that might impact the validity of the comparison.

Data source is very well explained.

--Thank you!

Please explain the decision to use only the surveys and not the lab results from the mobile examination center (MEC)

--The mobile examination center (MEC) data was not utilized in our study because it did not fit the scope of our research. Our study aimed to examine dietary intake among participants. Thus, we only used the 24-hour dietary recall data, which was collected using an online application. The MEC data included medical and biomarker data (e.g., blood pressure, blood glucose, BMI, etc.). Integration of MEC data was beyond the predetermined scope of our research. Future studies should explore differences in health status within non-Hispanic Black adults using the NHANES MEC data.

Data on nutrition and health are collected from participants by conducting a series of interviewer-administered questionnaires and physical examinations.

--This is correct.

How was the cut-off of < or > 10 years created? For FB<10yr n=167 across 11 years means approximately 15 per year.

--The decision to select 10 years as the cut-off was data driven. NHANES collects information on length of time in the U.S. in 10-year increments. There were not enough people in the higher categories of time (10-20 years, 20-30 years, etc.) to examine these groups separately. We decided to collapse the categories.

To further clarify, this is a dichotomous variable (<10 vs. ≥10). Because we do not have the exact time for each NHANES participants, we cannot calculate average time in the U.S. for all of the foreign-born participants.

FB Black adults who migrated less than 10 years ago (n = 167; 3.0%), FB Black adults who migrated greater than 97 10 years ago (n = 493; 7.2%), and US-born Black adults (n = 5,848; 89.8%).

--This is correct.

While it’s understandable the authors used the data available, something about the comparison to FB vs US born seems overly simplistic. 

--Lines 328-361: We appreciate your concern and acknowledge that the comparison between foreign-born (FB) and US-born individuals is simplistic. We do believe this fact warrants much more attention in the Discussion section of our manuscript. We revamped the limitations subsection and added a section header. We discussed this concern and provided our thoughts and recommendations for future research.

Despite the limitations of the NHANES data, our study did further illuminate dietary disparities within the non-Hispanic Black population in the US. We value your understanding of our research objectives and anticipate that our findings will provide valuable insights for the field.

If a foreign-born person immigrated with parents, are they the one’s maintaining their traditional diet? If a husband comes with his wife, is she the one shopping for or cooking traditional foods? Would a single person maintain a traditional diet?

--These are good questions. Unfortunately, our analysis and data source do not allow us to answer them. We do mention the importance of intergenerational relationships and family dynamics in the edits to the limitations subsection.

Given our current knowledge of the topic, we believe the maintenance of a traditional diet post-immigration varies among individuals and families, influenced by personal decisions, cultural values, and practical considerations.

Would a US born Black adult be aware of ways to achieve a diet similar to FB counterparts? Would that have been modeled in the home for them just because they share skin color?

--We can’t answer these questions given our data source and analysis. We can’t generalize the awareness and knowledge of all US-born non-Hispanic Black adults.

Are all US born Blacks direct descendants from FB countries represented?  What about the cultural/societal impact of diet related to mixed races and slavery?

--Again, we can’t answer these questions given our data source and analysis. NHANES does not have detailed ancestry data since participants do not provide their country of origin. Therefore, it is challenging to generalize that all US-born Blacks are direct descendants from the FB countries represented in the dataset.

NHANES data is primarily self-reported. However, it is important to recognize that the African American population in the US is highly diverse, and not all US-born Blacks trace their ancestry to countries represented in the African diaspora. Dietary practices can evolve over time, shaped by historical, cultural, and societal factors as well as individual preferences and available resources. In our edited limitations section. We acknowledge this complexity.

Results

For Table 2 – indicate within the table if there was a significant difference between groups.

--For Table 2, our goal was to simply provide estimates of the percentage of each group that met the nutrient intake recommendations. Given that we ran logistic regression models assessing differences in odds of meeting the recommendations between the 3 group (these results are in Table 3), we did not feel the need to show differences in Table 2 as well. We thought those results would have been repetitive given what is shown in Table 3.

Reviewer 2 Report

Dear Editors,

Thank you for the opprotunity to revise article „Differences in Nutrient Intake and Diet Quality among Non-2 Hispanic Black Adults by Place of Birth and Length of Time in 3 the United States”.

The article is very interesting. The study examined differences in nutrient intake and diet quality among non-Hispanic Black adults by place of birth and length of time in the US. Cross-sectional data from the National Health and Nutrition Examination Survey (2005-2016).

The article is well written. The introduction section is quiet short and might be improved.

The material and methods section is very detailed and there is no need to change anything.

The results are clearly demonstarted and the discussion is rigorous.

Thanks.

Author Response

Reviewer 2:

Dear Editors,

Thank you for the opportunity to revise article Differences in Nutrient Intake and Diet Quality among Non-2 Hispanic Black Adults by Place of Birth and Length of Time in the United States”.

The article is very interesting. The study examined differences in nutrient intake and diet quality among non-Hispanic Black adults by place of birth and length of time in the US. Cross-sectional data from the National Health and Nutrition Examination Survey (2005-2016).

The article is well written. The introduction section is quite short and might be improved.

--Thank you for your kind words and willingness to review our paper. We truly value your feedback and input. If you believe there are any important points that we may have overlooked in the introduction section, please do not hesitate to let us know. We are more than happy to consider any suggestions and make the necessary additions to ensure the completeness and accuracy of our work.

The material and methods section is very detailed and there is no need to change anything.

--We appreciate you providing us with this comment.

The results are clearly demonstrated, and the discussion is rigorous.

--We appreciate you providing us with this comment.

Thanks.